# Study on the Changes and Correlation of Microorganisms and Flavor in Different Processing Stages of Mianning Ham

**DOI:** 10.3390/foods13162587

**Published:** 2024-08-18

**Authors:** Yue Huang, Zhengli Wang, Ling Gan, Jiamin Zhang, Wei Wang, Lili Ji, Lin Chen

**Affiliations:** 1Key Laboratory of Meat Processing of Sichuan Province, Chengdu University, Chengdu 610106, China; 13684013852@163.com (Y.H.); jasminejjjjjj@163.com (J.Z.); wangwei8619@163.com (W.W.); jilili@cdu.edu.cn (L.J.); 2College of Agricultural Engineering and Food Science, Shandong University of Technology, Zibo 255000, China; 3College of Veterinary Medicine, Southwest University, Chongqing 402460, China; gl9089@swu.edu.cn

**Keywords:** Mianning ham, different processing stages, bacteria, fungi, volatile flavor, analysis of correlation

## Abstract

(1) Background: Mianning ham is a dry-cured ham from Southwest China, known for its distinct regional characteristics and delicious taste. It is particularly favored by consumers due to its safety, as no artificial nitrites are added during processing. The microbial changes during its different processing stages significantly influence the final product’s flavor. This study aims to investigate the changes in microbial communities and flavor compounds across the nine stages of Mianning ham production, from raw material preparation to salting, drying, fermentation, and maturation, using 16S and ITS sequencing, as well as solid-phase microextraction–gas chromatography–mass spectrometry (SPME-GC-MS). The goal is to explore the correlation between these changes and provide a basis for process improvement from the initial raw material preparation. (2) Results: The microbiota of Mianning ham across different processing stages mainly consisted of Proteobacteria, Firmicutes, and *Ascomycota*. A total of 324 volatile compounds were identified, of which 27 were key contributors to the ham’s flavor. Aldehydes contributed the most to flavor, with octanal, trans-2-nonenal, and trans, trans-2,4-decadienal being the most significant contributors at various stages. Mature Mianning ham, fermented for 1–3 years, exhibited fresh grass and earthy aromas, buttery and fatty flavors, and a distinctive roasted potato note. Correlation analysis revealed that *Cobetia* was the primary bacterial contributor to the main flavor compounds, especially prominent in the second year of fermentation. Among fungi, *Yamadazyma* and *Aspergillus* positively influenced several key aldehyde flavor compounds throughout the processing stages, significantly contributing to the flavor profile of Mianning ham. (3) Conclusions: Correlation analysis showed that the Mianning ham that matured for two years had the richest and most characteristic flavor. The positive and consistent impact of fungi on the ham’s flavor suggests that they may warrant further research and application in Mianning ham production. This provides a theoretical basis for improving the flavor quality and enhancing the market competitiveness of Mianning ham. One of the key features of Mianning ham is its substantial accumulation of hydrocarbons, which surpasses that of hams from other regions in China. A notable characteristic of Mianning ham processing is the absence of artificially added nitrites as antioxidants and color fixatives. Whether this absence is a contributing factor to the significant accumulation of hydrocarbons warrants further investigation.

## 1. Introduction

With the growth of the global economy, it is of great economic value to develop highvalue-added functional meat products, the meat industry is constantly innovating and developing [1].The unique and diverse flavor profile of dry-cured ham is attributed to complex reactions that are dependent on endogenous enzymes and microbial enzymatic activity [2,3]. However, the activity of endogenous enzymes tends to decrease during the processing and post-maturation stages of ham [4]. Many studies have demonstrated the contribution of microbial secondary metabolism and enzymatic reactions by microbial-derived enzymes to the formation of volatile organic compounds (VOCs) in ham [5,6,7], and there is also research indicating that microbes have a role in enhancing the flavor of fermented meat products [8,9]. The raw materials of dry-cured ham, added auxiliary materials, processing environment, and contact with hands are all significant sources of its microbiota [10]. During the various stages of dry-cured ham processing, the composition of microbial communities undergoes dynamic changes. Additionally, these microbes can directly participate in the formation of volatile flavor compounds in dry-cured ham through the catabolism of amino acids, metabolism of fatty acids, and degradation of carbohydrates [11].

Mianning ham, a traditional dry-cured ham from Mianning County, Sichuan Province, China, is an integral part of the local dietary culture. Unlike many other hams, Mianning ham emphasizes natural fermentation and health safety by not artificially adding nitrites during the processing and curing stages. It is made using high-quality pork from crossbreeds of Liangshan black pigs and superior white breeds like Landrace or Yorkshire. This approach not only gives the ham a unique flavor but also meets the modern consumer’s demand for natural and healthy foods, making it popular nationwide.

However, the changes in fermentation microorganisms and flavor compounds during the various processing stages of Mianning ham significantly affect its quality, leading to variability in the quality of different batches. Therefore, understanding the changes in microorganisms and flavor compounds at different stages of processing and the impact of microorganisms on these compounds is crucial. While there is extensive research on the correlation between microorganisms and flavor compounds in different types of dry-cured ham, no such studies have been conducted on Mianning ham. Mianning ham, recognized as a “Geographical Indication Product” by the Chinese government and boasting a health label for being free of added nitrites, requires theoretical research like the present paper to support broader promotion. Such studies could provide sellers with greater confidence and evidence to highlight the product’s advantages.

Moreover, many previous studies on dry-cured ham have focused primarily on bacterial communities, neglecting fungi. Research indicates that fungi can form a protective coating on the surface of fermented meat products, preventing moisture loss, reducing direct oxygen contact, and imparting unique flavors [12,13]. This study investigates the dynamic changes in bacterial and fungal communities during the processing of Mianning ham and their impact on flavor compounds, aiming to provide a theoretical foundation for improving its flavor quality and market competitiveness.

## 2. Materials and Methods

### 2.1. Sample Preparation and Sampling

The Jiuyuan Ham Factory in Mianning County, Liangshan Yi Autonomous Prefecture, Sichuan Province, was selected as the sampling site. Samples are taken randomly from the surface and inside of the hams, and fat and fascia were removed for high-throughput sequencing.

The ham samples are categorized according to the different processing stages of Mianning ham: raw ham material (HT1), initial curing stage sampled on the 7th day of salting (HT2), late curing stage sampled on the 35th day of salting (HT3), washing and sun-drying stage (HT4), 3 months into fermentation (HT5), 6 months into fermentation (HT6), 1 year of fermentation maturity (HT7), 2 years of fermentation maturity (HT8), and 3 years of fermentation maturity (HT9). Each stage was sampled three times in parallel. Samples were flash-frozen with liquid nitrogen and stored at −80 °C in the laboratory for subsequent microbial diversity and flavor compound analysis.

### 2.2. Genomic DNA Extraction and Polymerase Chain Reaction (PCR) Amplification

The genomic DNA of the samples was extracted with HiPure Soil DNA kit (according to the instructions), and the extracted DNA was purified by the ethanol precipitation method. Then, the purity and concentration of DNA were detected by NanoDrop microspectrophotometer. The sample size was 2 μL, and the concentration measurement range was 2–3000 ng/μL. The OD value of A260/A280 ratio should be between 1.8 and 2.0, and the ratio of A260/A230 should be about 2.2. The integrity of DNA was examined by 2% agarose gel electrophoresis. Genomic DNA with good quality should be a single band after agarose gel electrophoresis. After screening, the good-quality DNA was used for amplification of bacterial 16S rDNA and fungal ITS genomes. Specific primers 341 (F) (5′-CCtacGGGNGGCWGCAG-3′) and 806 (R) (5′-GGactachVGGGtatCTAat-3′) were used for PCR amplification of the V3–V4 region of bacteria [14]. Specific primers ITS3-KyO2 (F) (5′-GatgaagaacgyagyRAa-3′) and ITS4 (R) (5′-TCCTCCGCttattgatatGC-3′) were used for PCR amplification of fungal ITS2 region. The amplification system refers to the method of Chen et al. [4]. AMPure XP Beads were used to purify the second round of amplification products. Then, the PCR amplification products were cut and recovered, and Qubit3.0 was used for quantification.

### 2.3. Illumina Miseq Sequencing

Amplicons were collected from 2% agarose gels, purified using the AxyPrep DNA Gel Extraction kit (Axygen Biotechnology Co., Ltd., Zhejiang, Hangzhou, China) according to the manufacturer’s instructions, and quantified using the ABI StepOnePlus real-time PCR system. The purified amplification products were mixed in equal volume and connected to sequencing connectors to construct sequencing libraries, and then double-ended sequencing was performed on the Illumina platform according to the PE 300 mode of Miseq.

To ensure high-quality sequencing data, we used fastp (version 0.18.0, OpenGene, https://github.com/OpenGene/fastp, accessed on on 6 November 2022) [14,15] to filter the raw data. The specific operations included removing reads containing unknown nucleotides (N) ≥ 10% and removing reads with a Phred quality score ≤ 20 for ≥50% of the bases. Reads containing connectors were deleted, and clean reads obtained after filtering were used for assembly analysis. Paired-end clean reads were merged as raw tags using FLSAH (version 1.2.11) [16] with a minimum overlap of 10 bp and mismatch error rates of 2%. According to the filtering conditions presented in [17], Clean Tags were obtained by filtering low-quality tags. Refer to Caporaso et al. [18] for the tags quality control process. Clean Tags were clustered into OTUs (operational taxa) by ≥97% similarity using the UPARSE (version 9.2.64) [19] process. The UCHIME algorithm [20] was used for tag chimera inspection (for 16S sequencing analysis). Effective tags were obtained after filtering, and clustering was carried out under 97% similarity. The tag sequence with the highest abundance was selected as the representative sequence of each OTU. The representative sequence set of OTUs were classified into organisms by a naive Bayesian model using RDP classifier (version 2.2) [21], with the confidence threshold value of 0.8. Bacterial 16S was compared with SILVA [22] database, and fungal ITS was compared with UNITE [23] database to annotate species classification information.

### 2.4. Processing and Calculation of Volatile Flavor Compounds in Samples

The samples of Mianning ham in different processing stages were finely chopped, and 3.00 g was accurately weighed into a 15 mL headspace bottle. A measure of 1 μL of 2,4,6-trimethylpyridine standard solution was added to the headspace bottle as an internal standard, and then refrigerated for later use. The detection method of volatile flavor compounds was referred to the method of Chen et al. [4].

The absolute contents of volatile flavor compounds were calculated according to Equation (1):(1)$$C=\frac{C_{1}\times V\times A}{A_{1}\times m}$$
where C (μg/kg) is the absolute content of each volatile flavor compound; *C*_1_ (μg/ μL) is the mass concentration of internal standard; *A* is the peak area of each volatile flavor compound; *A*_1_ is the peak area of internal standard; *m* (g) is the sample mass; *V* (μL) is the added volume of the internal standard.

The key volatile flavor compounds were calculated by the odor activity value (OAV) method according to Equation (2):(2)$$\mathrm{OAV}=\frac{B_{1}}{B_{2}}$$
where *B*_1_ (μg/kg) is the content of compound in the sample; *B*_2_ (μg/kg) is the odor threshold of the compound.

## 3. Results

### 3.1. Analysis of Microbial Diversity in Different Processing Stages

#### 3.1.1. Microbial Sequencing Data

IIIumina Miseq high-throughput sequencing technology was used to obtain raw reads from samples at different processing stages of Mianning ham. After that, low-quality reads were filtered, and then assembled and re-filtered to obtain effective sequences. A total of 310,715 effective sequences were obtained in the V3–V4 region of bacterial 16S, and 3,347,626 effective sequences were obtained in the ITS2 region of fungal ITS. The original data have been submitted to the SRA database of NCBI, with the following project key number: PRJNA679127.

#### 3.1.2. Microbial Diversity Analysis

The dilution curve can evaluate whether the sequencing quantity was sufficient and indirectly reflect the richness of microbial species in the sample. When the dilution curve flattens out or reaches a plateau, it can be considered that the increase in sequencing depth does not affect the species diversity, indicating that the sequencing quantity was sufficient. At the same time, the difference of α-diversity structure between the different samples could be reflected by a comparison of the height of the curves between early and late reaching of a plateau. From the x-axis of Figure 1, it can be observed that the dilution curves for bacteria and fungi at various stages have leveled off, indicating that the sequencing depth in this experiment is sufficient. In Figure 1A, the height of the curves shows that bacterial diversity is highest at the HT7 stage and lowest at the HT9 stage. In Figure 1B, it is evident that fungal diversity is highest at the HT4 stage and lowest at the HT9 stage.

The Alpha diversity index was shown in Table 1 and Table 2. The microbial sequence coverage of Mianning ham at different processing stages was ≥0.99. This indicates that the amount of sequencing data was reasonable and sufficient to reflect the microbial information contained in different processing stages of Mianning ham. The dilution curve evaluates the ability of each sample to reflect the bacterial diversity in the sample at the current sequencing depth [24]. The sparse curves of the nine different processing stages all reached a plateau, indicating that the sequencing depth was sufficient, and the sequencing results could fully cover the bacterial information contained in the samples (Figure 1). Operational taxonomic units (OTUs) were clustered and annotated with 97% consistency. A total of 11,212 OTUs were obtained by bacterial 16S division and 3586 OTUs were obtained by fungal ITS division. The number of OTUs of bacteria was higher than that of fungi, indicating that the diversity of bacteria in Mianning ham was richer. The Shannon, Chao, and Ace indices were the highest in the washing and drying stages, while the Shannon, Chao, and Ace indices were the lowest in the 3-year stage, indicating that the microbial species and abundance of Mianning ham in the washing and drying stages were the highest, and those in the 3-year stage were the lowest. However, a study of Xuanwei ham, another characteristic ham in Southwest China, yielded the opposite result, i.e., the microbial species and abundance of Xuanwei ham in the third year of fermentation were found to be the highest [25].

#### 3.1.3. Analysis of Microbial Community Differences

The species distribution of microbial communities in different habitats has a certain degree of similarity and specificity. In order to understand the differences in the OTUs between samples at different processing stages of Mianning ham, we can carried out a Venn diagram analysis according to the abundance information of OTUs. The goal of this was to understand the common or unique information of OTUs between samples in different processing stages. If the mean number of OTU tags in a group was greater than 1, then the OTU exists in the group. In the Venn diagram, different colored graphs represent different processing stages, and the overlapping numbers between the different colors are the common OTU numbers of hams in different processing stages.

It can be seen from Figure 2 that the distribution of microbial communities in samples representing different processing stages of Mianning ham exhibits distinctive characteristics and commonalities. In the nine different processing stages, the total number of bacteria OTUs was 109, and the total number of fungi OTUs was 53. 

In the study of Nuodeng ham, a characteristic ham in Southwest China, it was found that the total number of OTUs of bacteria in the curing period, the air-drying period, the fermentation period, and the mature period was 61 [26]. There were 840 unique bacterial OTUs at the 1-year stage, which was the largest number. Fungi had the highest number of OTUs at the 3-year stage, with 81 OTUs. In the study of microbial diversity in five characteristic hams in Southwest China, Sanchuan ham, Saba ham, Xuanwei ham, Laowo ham, and Nuodeng ham, researchers have found that the numbers of unique bacterial OTUs in the five hams at 1 year of fermentation were 476, 831, 1241, 1359, and 998, respectively; the numbers of fungal OTUs were 47, 42, 58, 33, and 28, respectively [27]. Compared with other hams, the number of OTUs of bacteria and fungi in Mianning ham at the same stage of fermentation for 1 year was relatively low.

During the processing of Mianning ham, the number of bacterial OTUs showed a downward trend from the raw material to the end of the curing stage, indicating that salt inhibited the growth and reproduction of bacteria during the curing process of ham. In the later stage of ham washing and drying, the bacteria had a small degree of growth because the salt on the surface of the ham was washed away. As the fermentation progresses, the rapid decrease in moisture affects the growth of bacteria, the number of bacteria generally showed an upward trend. From the raw materials stage to the washing and drying stages, fungi and bacteria have the same growth trend. In the fermentation stage, with a rapid loss of water, the number of fungi first decreased and then increased. In the HT5 stage, as the ham enters the fermentation chamber and begins fermenting, environmental changes may lead to a reduction in fungi. However, during the subsequent fermentation stages, the fungal OTUs show slight fluctuations but generally exhibit an upward trend.

Principal component analysis (PCA) was carried out based on the abundance information for the OTUs. The closer the distance in the PCA map, the more similar the sample composition is. According to Figure 3A, the raw material stage overlaps with the curing stage, and the washing and drying stage overlaps with the 1-year stage, indicating that the bacterial composition they cluster around, respectively, is very similar. The 3-month and 6-month fermentation stages were clustered in the third quadrant, indicating that the bacterial composition changed during the fermentation process. However, the 3-year ham was not in the same quadrant as the other stages, indicating that the bacterial composition of the 3-year stage was not similar to the other stages. As can be seen from Figure 3B, fungi cluster in the third quadrant in the raw material stage, and with the processing of Mianning ham, there is a trend of gradual convergence in Figure 3B, indicating that the composition of fungi becomes more and more similar during the processing of Mianning ham. There was an overlap between the 3-month fermentation stage and the 6-month fermentation stage, indicating that the fungal compositions of the two stages were very similar. The distance between the 1-year and 3-year hams and the whole hams was large, indicating that the fungal compositions of the 1-year and 3-year hams were not very similar to the fungal compositions of other stages. In conclusion, the microbial composition of Mianning ham will change greatly and form a unique microbial composition in the later stages of maturation.

#### 3.1.4. Analysis of Microbial Community Composition

The stacking diagram can be used to visually display the microbial species composition in different processing stages of ham, find the changing trend of microbial population in different processing stages, and evaluate the species with the largest or most stable changes and dominant species. The species ranked top 10 in the mean abundance of all samples were selected and displayed in detail. Other known species were classified into others, while unknown species were marked as unclassified.

As shown in Figure 4, the dominant bacterial phylum in stages HT1–4 and HT8 is Proteobacteria, with relative abundances of 51.36%, 56.71%, 65.62%, 55.74%, and 50.80%, respectively. During the hanging fermentation stages, where the ham is in full contact with the air, the dominant bacterial phylum shifts to Firmicutes, with relative abundances of 48.74%, 88.73%, 38.36%, and 98.74% for stages HT5, HT6, HT7, and HT9, respectively.

Yang et al. demonstrated that Actinobacteria have an inhibitory effect on Proteobacteria in dry-cured hams [28]. However, in this study, during the stages where Proteobacteria were dominant, the numbers of Proteobacteria remained high, suggesting that there might be other factors influencing their growth. Studies have shown that Firmicutes in cured meat products are closely related to carbohydrate metabolism, while Proteobacteria are closely associated with amino acid and lipid metabolism [29]. However, during the processing of Mianning ham, various ongoing biochemical reactions and changes in environmental conditions result in a dynamic alteration of the ham’s microbiota. In stages HT7–9, Proteobacteria and Firmicutes exhibit competitive growth, alternating as the dominant phylum. By stage HT9, in the third year of ham maturation, Firmicutes almost completely dominate. This dominance is likely due to Firmicutes’ ability to thrive in the low moisture and high salt concentration environment characteristic of the third-year stage of ham maturation. Similarly, studies on Nuodeng ham from Yunnan, China, have found comparable results. During its curing and air-drying stages, the dominant phylum is also Proteobacteria. From six months to a year of fermentation, the dominant phylum in both hams is Firmicutes [26]. 

During the HT1–2 stages, the dominant bacterial genus was *Brochothrix*, with relative abundances of 31.43% and 20.93%, respectively. *Brochothrix* is a common spoilage microorganism with some tolerance to salt, but it appears in large quantities only during the HT1–2 stages. Due to its sensitivity to high temperatures [30], its presence significantly decreases after the HT4 washing and sun-drying stage. In the HT3, the dominant bacterial genus was *Pseudomonas*, with a relative abundance of 31.01%. This could be due to the curing stage being conducted at 4℃ in a cold storage room for 40 days. The most common species in this genus, *Pseudomonas fluorescens,* has strong low-temperature tolerance [31]. During the HT4, the dominant bacterial genus was *Acinetobacter*, with a relative abundance of 9.46%. This is likely because the exposure to air and sunlight provided the necessary oxygen and suitable temperature for *Acinetobacter* growth [32]. 

*Staphylococcus* was the dominant genus during the HT5–6 stages of Mianning ham, with its relative abundance showing a rapid increasing trend. In Nuodeng ham, *Staphylococcus* was also the most abundant genus throughout the entire processing period [26]. Similar findings have been reported for Jinhua ham [33]. *Staphylococcus* is known to produce proteases and lipases, which are beneficial for the generation of flavor compounds in ham, especially aldehydes [33]. This is supported by Figure 5, which shows a significant increase in aldehyde compounds during the HT5–HT6 stages. After the substantial decrease in *Staphylococcus* abundance, the number of aldehyde compounds also stabilized.

As fermentation continued, the moisture content in the ham decreased, while the salt content increased. Consequently, the dominant genus in the HT7 stage was *Psychrobacter*, with a relative abundance of 11.90%. In the HT8 2-year stage, *Cobetia* was the dominant genus, with a relative abundance of 12.19%. Both genera belong to the Proteobacteria phylum and are halotolerant, consistent with previous findings [34,35]. *Psychrobacter*, being more psychrophilic, might have decreased due to environmental temperature changes between the 1st and 2nd year of fermentation. In the HT9 stage, the dominant genus was *Lentibacillus* [36], with a relative abundance of 94.77%. As a member of the Firmicutes phylum, *Lentibacillus* exhibits strong salt tolerance [37]. Although it has been found in fermented seafood, its presence in fermented meat products is relatively rare. It is speculated that its origin could be the salt used during the curing process [38].

Fungi form a protective layer on the surface of fermented meat products, preventing moisture loss, reducing direct oxygen exposure, and imparting unique flavors [12,13]. The results indicate that the dominant fungal phylum across all stages of Mianning ham processing is *Ascomycota*, with its relative abundance consistently increasing throughout the stages: 41.31%, 48.27%, 81.32%, 71.54%, 75.33%, 93.59%, 99.38%, 82.16%, and 99.78%. In contrast, the dominant fungal phyla in the one-year-old or older Sanchuan, Saba, Xuanwei, Laowo, and Nuodeng hams include *Ascomycota*, *Florophyta*, and *Basidiomycota*. Notably, Xuanwei ham [39], Laowo ham, and Sabaa ham [40] share the same dominant fungal phylum, *Ascomycota*, with Mianning ham [27].

At the genus level, the dominant fungal genera in Mianing ham at different processing stages are *Aspergillus*, *Yamadazyma*, and *Candida*. The relative abundance of *Aspergillus* generally increases throughout the various processing stages, with a rapid increase observed during the HT4–5 stages and a positive correlation during the other stages. *Aspergillus* plays a crucial role in food fermentation [41], producing proteases and lipases that contribute to the flavor formation of the ham. *Yamadazyma* shows higher relative abundance during the late curing stage, 3 months of fermentation, and 6 months of fermentation. The relative abundance of *Yamadazyma* decreases in the HT4 stage due to washing, while *Aspergillus* experiences rapid growth and becomes dominant in the HT7 stage. *Yamadazyma* is mainly present in the later curing and fermentation stages, while *Aspergillus* is more prevalent in the maturation stages. Similarly, in the fermentation process of Xuan’en ham, *Aspergillus* was also found to be the dominant fungal genus. Further studies indicated a species succession at the species level, with *Aspergillus* cibarius being dominant in the early and middle fermentation stages (1 to 3 months) and *Aspergillus* penicillioides becoming dominant in the late fermentation stage (one and a half years) [42].

### 3.2. Analysis of Volatile Flavor Compounds

#### 3.2.1. Analysis of Volatile Flavor Compounds in Different Processing Stages

In different processing stages of Mianning ham, a total of 324 volatile compounds were identified using SPME-GC-MS technology. These include 41 aldehydes, 25 ketones, 28 acids, 33 esters, 68 alcohols, 93 hydrocarbons, and 36 other compounds, as detailed in Appendix A. Compared to hams produced in other parts of southwestern China, Mianning ham has a significantly higher number of volatile flavor compounds. For instance, in Dahe black pig ham produced in Yunnan, China, only 137 volatile flavor compounds were identified during its 3 months–2 years fermentation period [43]. Similarly, San Chuan ham from Yunnan, China, had only 58 volatile flavor compounds identified during its 6 months–1 year fermentation period [44].

As shown in Figure 5, the 3-year-old Mianning ham had the highest number of flavor compounds, totaling 177, followed by the 1-year and 2-year hams, with 167 and 133 compounds, respectively. In the raw material, early curing, late curing, washing, drying, 3-month fermentation, and 6-month fermentation stages, aldehydes were the most abundant flavor compounds. In the 1-year, 2-year, and 3-year fermentation stages, hydrocarbons became the most abundant. The raw material stage had the fewest volatile flavor compounds, with only 10 compounds, and no ketones, acids, or hydrocarbons. As Mianning ham matures, the overall number of volatile flavor compounds increases, with the 6-month fermentation stage showing a substantial increase and the 3-year stage having the highest number.

In the different processing stages of Mianning ham, hydrocarbons are the most abundant volatile flavor compounds, particularly in the HT7 and HT9 stages, where the number of hydrocarbon compounds increases rapidly. Studies have shown that hydrocarbons mainly originate from the oxidation of fats and the unsaponifiable components in feed products, with aliphatic hydrocarbons being produced by lipid oxidation degradation [45]. However, due to their high flavor threshold, hydrocarbons are generally considered to contribute little to the overall flavor of ham [46]. The absence of added nitrites is a significant processing characteristic of Mianning ham. Since nitrites have strong antioxidant properties, whether their absence is related to the substantial accumulation of hydrocarbons in Mianning ham warrants further investigation. The next most abundant compounds are alcohols and aldehydes. This is different from other Chinese hams, such as Nuodeng ham [47], Sanchuan ham [44], and Dahe black pig ham [43], where aldehydes and alcohols are the most abundant volatile compounds, while hydrocarbons are generally fewer. Ketones, acids, alcohols, esters, hydrocarbons, and other compounds are primarily distributed in stages after 6 months of fermentation.

The total absolute content of all volatile flavor compounds in Mianning ham during different processing stages is 87,872.22 μg/kg. Aldehydes have the highest total absolute content, reaching 48,547.65 μg/kg. As seen in Table 3, the absolute content of aldehydes gradually increases from the HT1 to HT4 stages, peaking at 10,527.94 μg/kg during the HT4 stage, indicating that the washing and drying process during this period may lead to fat oxidation, resulting in the accumulation of a large number of aldehydes. Alcohols have the second highest absolute content at 12,915.65 μg/kg, showing an initial increase followed by a decrease across different processing stages. These include straight-chain alcohols, branched-chain alcohols, enols, aromatic alcohols, and diols, most of which are also the oxidation products of lipids [46]. Notably, except for ketones and lipid compounds, all other flavor compounds have the highest absolute content during the HT4 washing and drying stages, where the oxidation rates of proteins and fats in Mianning ham are relatively high, significantly contributing to the ham’s flavor.

#### 3.2.2. Key Volatile Flavor Compounds 

Odor activity value (OAV) can identify the major contributing volatile flavor compounds. Using the OAV method, the key volatile flavor compounds in Mianning ham during different processing stages were identified, as shown in Table 4. A total of 27 key volatile flavor compounds were identified across different processing stages of Mianning ham. When OAV ≥ 1, the contribution of volatile flavor compounds is directly proportional to the OAV; the larger the OAV, the greater the contribution of the substance to the ham’s flavor. When 0 ≤ OAV < 1, the volatile flavor compound is considered to have some influence on the flavor, but the effect is minor, and it is not a key substance. Among the 27 key compounds, aldehydes stand out in terms of contribution. Aldehydes have a low threshold and a wide variety, making them significantly influential in the flavor of Mianning ham. Particularly, compounds such as octanal, trans-2-nonenal, and trans, trans-2,4-decadienal contribute greatly to the flavor. Alcohols also have a notable contribution to the flavor, with 1-octen-3-ol being particularly significant.

Considering that Mianning ham generally needs to ferment for at least one year to be considered mature and market-ready, a summary of the key flavor compounds in the matured ham from stages HT7–9 is presented. According to the results in Table 4, aldehydes are the most significant contributors to the flavor profile. Among them, octanal stands out as the most influential, with hexanal and nonanal making comparatively minor contributions. Research indicates that octanal imparts a fresh, grassy aroma [48], while hexanal, a major oxidation product in dry-cured meats, typically results from the degradation of linoleic acid [49] and provides green apple and vegetable notes [50,51]. Notably, several less common but highly impactful aldehydes were found in mature Mianning ham: trans-2-nonenal and trans,trans-2,4-decadienal. These unsaturated aldehydes are significant contributors, just below octanal, and are known for enhancing fatty flavors [52]. Another rare unsaturated aldehyde, 3-methylthio-propanal, imparts a cooked potato flavor [53]. These compounds have the most substantial influence on the flavor of Mianning ham during the second year of fermentation, indicating that the ham develops distinctive buttery, fatty, and cooked potato notes at this stage. Alcohols also contribute to the flavor, though to a lesser extent. Among them, 1-octen-3-ol is present at all stages of maturation, providing sweet, earthy, and herbal aromas (https://pubchem.ncbi.nlm.nih.gov/compound/1-Octen-3-OL, accessed on 20 July 2024). Previous studies have identified the characteristic flavor compounds of Jinhua ham as hexanal and nonanal [33], Dahe Wu pig ham and Xuanwei ham as octanal and nonanal [43], and Xuanwei ham as hexanal and heptanal [54]. While Mianning ham shares some key flavor compounds with these hams, the overall influence of hexanal and nonanal is relatively minor, highlighting Mianning ham’s unique and distinctive flavor profile.

#### 3.2.3. Correlation Analysis

The Pearson correlation analysis was conducted on the dominant microorganisms and key volatile flavor compounds in different processing stages of Mianning ham, as shown in Figure 6.

From the perspective of microbial phyla, although Proteobacteria and Firmicutes are often negatively correlated with many key flavor compounds, Proteobacteria show a certain positive correlation with most of the significant aldehyde compounds contributing to *Mianning* ham’s flavor, except for 3-methylthio-propanal. In contrast, Firmicutes do not have a significant relationship with the key contributing aldehydes but do exhibit some positive correlations with a few alcohols, phenols, and ketones. Given that Firmicutes are predominantly abundant in HT9 (the third-year stage), and this leads to the lowest OAV values for all key flavor compounds at this stage, it is likely that Firmicutes significantly diminish the unique flavor of *Mianning* ham during this period. This is supported by the lowest absolute content of flavor compounds at this stage, as shown in Table 3, which might reduce consumer purchasing intent. *Ascomycota*, however, is the fungal phylum most positively correlated with various key flavor compounds, potentially enhancing the diversity of flavors in mature *Mianning* ham. Unfortunately, it seems to have a negligible effect on several aldehydes crucial for the flavor of *Mianning* ham. Nevertheless, *Ascomycota* does show a positive correlation with 3-methylthio-propanal, which is important for the flavor of mature *Mianning* ham.

From the heatmap analysis, it can be observed that, during the HT1–3 stages, dominant bacteria such as *Brochothrix* and *Pseudomonas* show minimal significant relationships with the production of key flavor compounds, only exhibiting a certain positive correlation with hexanal. In the HT4 stage, the dominant bacteria *Acinetobacter* has a notable positive correlation with the distinctive flavor compound octanal in *Mianning* ham. Subsequently, during the HT5–6 stages, *Staphylococcus*, which becomes the dominant bacterium, demonstrates positive correlations with most key flavor compounds. This suggests that *Staphylococcus* may play a crucial role in the accumulation of diverse flavor compounds during these stages. In the mature HT7–9 stages, *Psychrobacter* and *Lentibacillus*, dominant in HT7 and HT9, do not show very positive impacts on key flavor compounds. However, *Cobetia*, dominant in HT8, is identified as the most significant bacterial species influencing key flavor compounds in *Mianning* ham. It exhibits the most substantial positive correlations with various key flavor compounds, including the most prominent flavor compound, octanal. *Cobetia* also shows significant positive correlations with other major flavor contributors such as 3-methylthio-propanal, trans-2-nonenal, and (E,E)-2,4-decadienal. This indicates that *Cobetia* plays a crucial role in developing the richest and most intense characteristic flavors in *Mianning* ham during HT8 stage.

Fungal species show little variation in dominance across different processing stages. During the HT1–2 stages, the dominant fungus *Candida* does not have a significant positive impact on key flavor compounds. However, *Yamadazyma*, which holds an important position in several stages, emerges as the most influential fungus for producing 3-methylthio-propanal, trans-2-nonenal, and trans,trans-2,4-nonadienal. From the HT4 stage onward, *Aspergillus* becomes the dominant fungus and demonstrates clear positive correlations with various key flavor compounds and the most important aldehydes in *Mianning* ham. Given its consistent dominance across all stages of processing, *Aspergillus* likely plays a crucial role in the production of key flavor compounds in *Mianning* ham.

## 4. Conclusions

In *Mianning* ham, bacteria at different processing stages are primarily composed of Proteobacteria and Firmicutes, while fungi are predominantly *Ascomycota*. The microbial community continuously evolves throughout the processing stages. At the genus level, the dominant bacteria include *Staphylococcus*, *Brochothrix*, *Pseudomonas*, *Psychrobacter*, *Acinetobacter*, *Cobetia*, and *Lentibacillus*. The key fungal genera are *Aspergillus*, *Yamadazyma*, and *Candida*.

A total of 324 volatile compounds were identified in *Mianning* ham across different processing stages, with hydrocarbons being the most diverse and abundant. The accumulation of hydrocarbons in Mianning ham, which surpasses that of hams from other regions in China, highlights a unique characteristic that warrants further investigation. Of the 27 key flavor compounds analyzed, aldehydes are the primary contributors to flavor. Notably, compounds such as hexanal, trans-2-nonenal, and trans,trans-2,4-decadienal are the most significant flavor contributors across all stages and in mature *Mianning* ham. These aldehydes impart fresh, grassy, buttery, and fatty notes to the ham. Additionally, the unique compound 3-methylthio-propanal, present in the mature stages, provides a roasted potato flavor. Among alcohols, 1-octen-3-ol contributes a distinctive sweet, earthy, and herbal note to the ham.

According to the correlation analysis, key flavor compounds in *Mianning* ham begin to accumulate significantly after the HT4 washing and drying stage. During these stages, the dominant bacteria, *Acinetobacter* and *Staphylococcus*, play crucial roles in this accumulation. In the HT7–9 maturity stages, *Psychrobacter*, which is dominant in HT7, seems to have a minimal positive effect on flavor. However, *Cobetia*, dominant in HT8, is identified as the most influential bacterium regarding key flavor compounds, showing a strong positive correlation with many critical flavor compounds and aldehydes that significantly contribute to flavor. This suggests that *Mianning* ham in the second year of fermentation exhibits the richest and most characteristic flavors. In contrast, *Lentibacillus* in HT9 appears to negatively affect the expression of flavor compounds. Given the low moisture and high salt content in the third year, the second year may be the optimal period for selling *Mianning* ham.

It is also noteworthy that fungi play a significant role in the flavor expression of *Mianning* ham. *Yamadazyma* and *Aspergillus* have a strong positive impact on various key aldehyde flavor compounds, second only to *Cobetia*. The relatively stable dominance of these fungal species across different stages suggests that their consistent and positive contribution could warrant further research and application in *Mianning* ham production.

## Figures and Tables

**Figure 1 foods-13-02587-f001:**
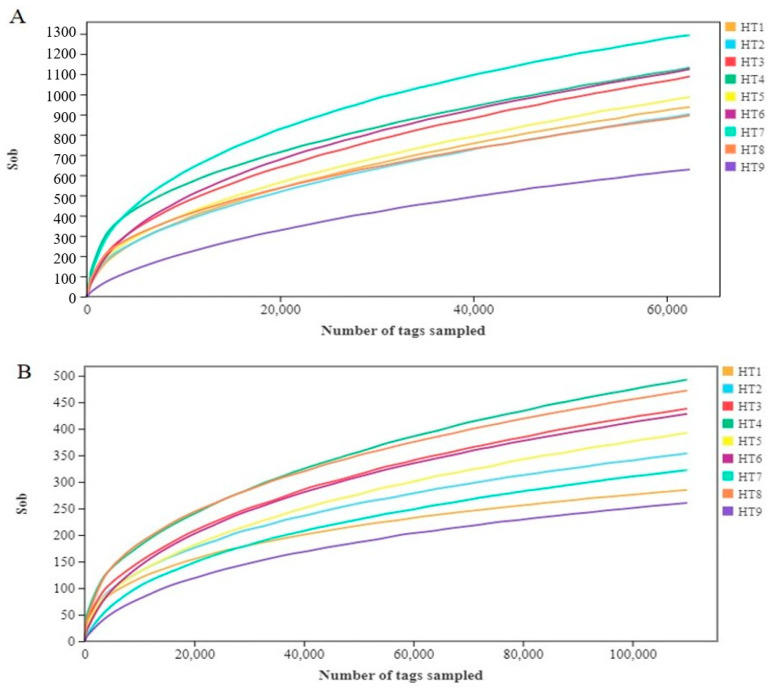
Richness sparse curve of sobs (observed OTUs) based on sample sequence. Note: (**A**) sparse curve of bacterial richness; (**B**) sparse curve of fungal richness.

**Figure 2 foods-13-02587-f002:**
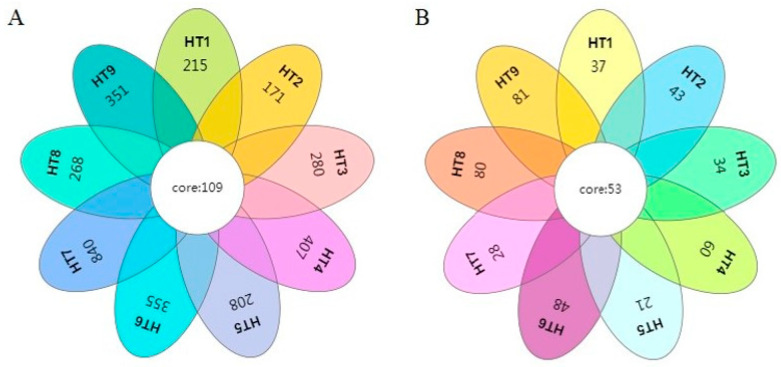
Venn diagram based on OTUs. Note: (**A**) bacterial OTU Venn diagram; (**B**) fungal OTU Venn diagram.

**Figure 3 foods-13-02587-f003:**
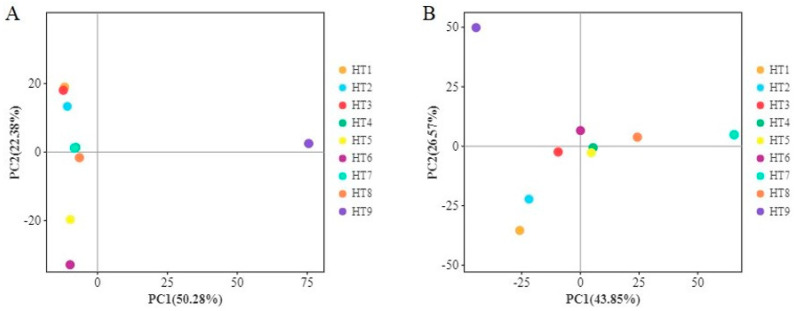
Score chart of principal component analysis based on OTUs. Note: (**A**) bacterial score chart; (**B**) fungal score chart.

**Figure 4 foods-13-02587-f004:**
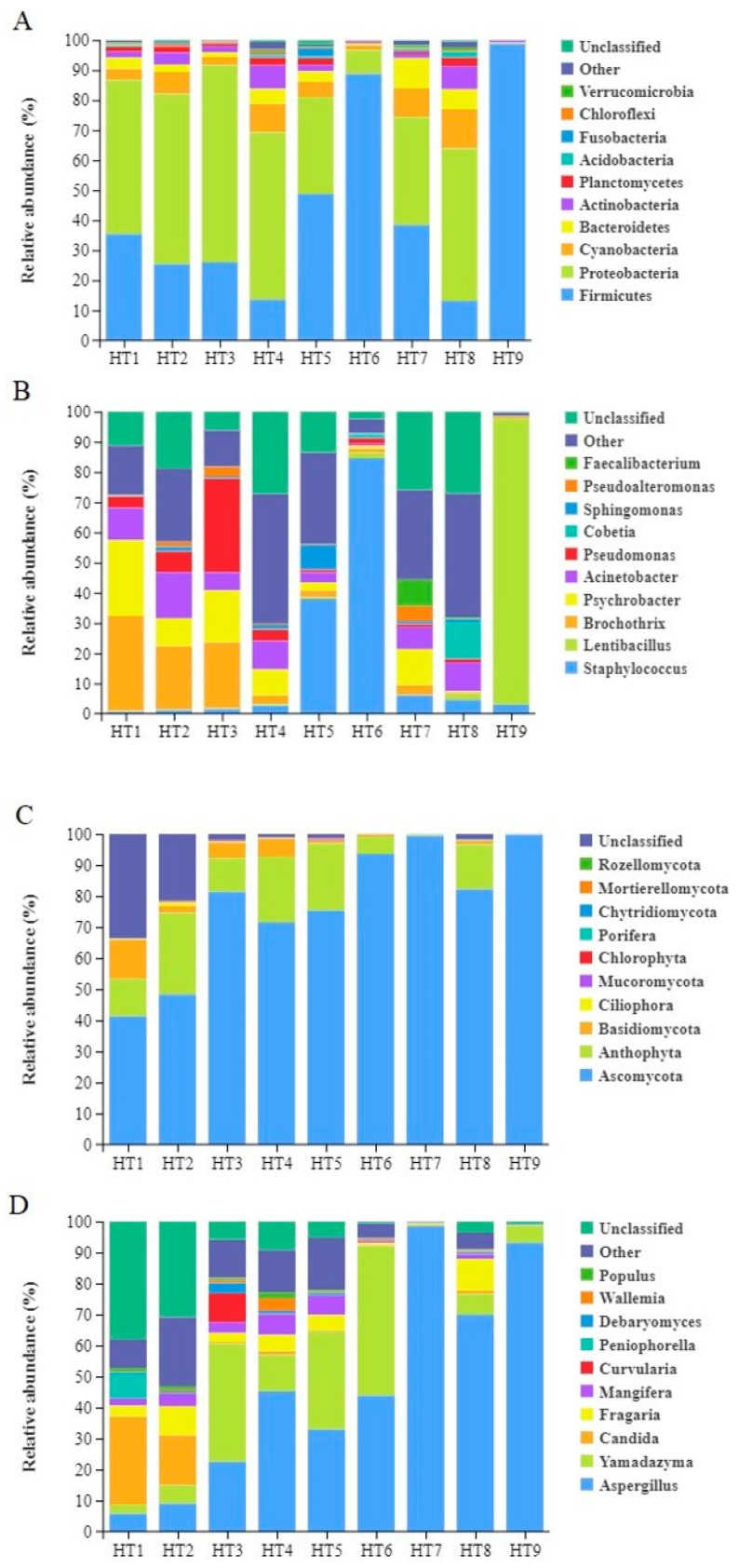
Stacking diagram of species relative abundance of microorganisms at phylum and genus levels. Notes: (**A**) stacked diagram of bacteria at phylum level; (**B**) stacked diagram of bacteria at genus level; (**C**) stacked diagram of fungi at phylum level; (**D**) stacked diagram of fungi at genus level.

**Figure 5 foods-13-02587-f005:**
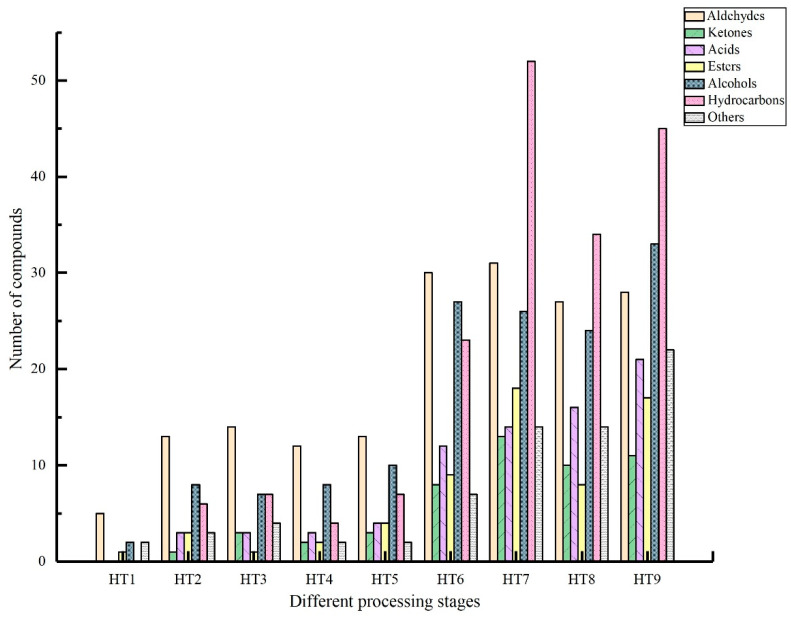
Changes of volatile compounds in Mianning ham at different processing stages.

**Figure 6 foods-13-02587-f006:**
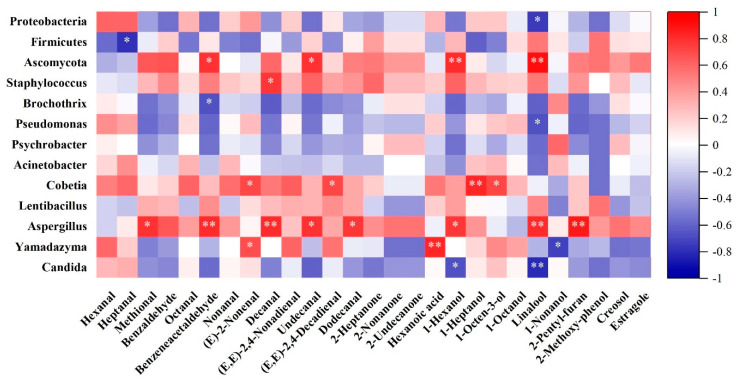
Heat map of correlation analysis between dominant microorganisms and key flavor compounds. Note: in the correlation heat map, the top three rows show the correlation between dominant phyla and key volatile compounds, and the bottom ten rows show the correlation between dominant genera and key volatile compounds; * significant correlation; ** extremely significant correlation.

**Table 1 foods-13-02587-t001:** Bacterial 16S rDNA alpha diversity index.

Group	Observed OTUs	Shannon Index	Simpson	Chao Index	Ace Index	Good’s Coverage
HT1	1198.00 ± 74.67	4.61 ± 1.96	0.83 ± 0.18	1983.71 ± 114.59	2048.28 ± 136.15	0.99 ± 0.00
HT2	1166.00 ± 84.12	5.28 ± 1.93	0.87 ± 0.17	1896.46 ± 77.46	2022.61 ± 102.18	1.00 ± 0.00
HT3	1400.00 ± 78.48	4.78 ± 0.93	0.86 ± 0.09	2299.60 ± 132.60	2369.53 ± 102.40	0.99 ± 0.00
HT4	1430.67 ± 60.87	7.22 ± 0.11	0.98 ± 0.00	2339.85 ± 80.57	2469.03 ± 47.70	0.99 ± 0.00
HT5	1264.67 ± 128.59	5.27 ± 0.49	0.90 ± 0.05	2113.66 ± 197.84	2322.46 ± 170.39	0.99 ± 0.00
HT6	1355.33 ± 20.21	4.30 ± 0.04	0.78 ± 0.00	2115.79 ± 49.11	2177.94 ± 33.50	0.99 ± 0.00
HT7	1404.00 ± 277.70	6.35 ± 0.50	0.95 ± 0.02	2006.18 ± 219.88	2125.99 ± 235.03	0.99 ± 0.00
HT8	1160.67 ± 98.57	6.26 ± 0.40	0.96 ± 0.03	1924.93 ± 127.87	2056.64 ± 126.76	1.00 ± 0.00
HT9	832.67 ± 34.53	1.41 ± 0.19	0.28 ± 0.05	1293.89 ± 20.65	1411.36 ± 22.90	1.00 ± 0.00

**Table 2 foods-13-02587-t002:** Fungal ITS diversity index.

Group	Observed OTUs	Shannon Index	Simpson	Chao Index	Ace Index	Good’s Coverage
HT1	290.00 ± 117.48	2.84 ± 1.48	0.71 ± 0.17	418.71 ± 206.38	409.47 ± 209.09	1.00 ± 0.00
HT2	371.00 ± 116.50	3.87 ± 1.86	0.82 ± 0.21	608.01 ± 179.59	562.92 ± 196.29	1.00 ± 0.00
HT3	453.67 ± 35.50	3.56 ± 0.58	0.80 ± 0.05	681.33 ± 81.51	701.86 ± 77.60	1.00 ± 0.00
HT4	517.00 ± 44.31	4.28 ± 0.34	0.87 ± 0.04	736.83 ± 39.29	780.05 ± 41.75	1.00 ± 0.00
HT5	416.00 ± 25.12	3.49 ± 0.22	0.81 ± 0.02	637.21 ± 27.76	671.17 ± 42.81	1.00 ± 0.00
HT6	443.67 ± 6.43	2.29 ± 0.03	0.67 ± 0.01	687.36 ± 40.14	666.42 ± 20.26	1.00 ± 0.00
HT7	335.33 ± 38.03	0.57 ± 0.02	0.11 ± 0.01	534.35 ± 23.44	526.25 ± 60.44	1.00 ± 0.00
HT8	493.67 ± 37.42	2.92 ± 0.34	0.69 ± 0.07	726.49 ± 31.17	741.44 ± 51.36	1.00 ± 0.00
HT9	266.00 ± 31.43	1.22 ± 0.20	0.38 ± 0.11	378.32 ± 44.55	387.73 ± 55.90	1.00 ± 0.00

**Table 3 foods-13-02587-t003:** Volatile flavor compounds content statistics of Mianning ham at different processing stages.

Compounds	Absolute Content (μg/kg)									
	HT1	HT2	HT3	HT4	HT5	HT6	HT7	HT8	HT9	Total
Aldehydes	101.37	6224.53	7127.77	10,527.94	5487.13	4693.04	4977.81	7101.65	2306.42	48,547.65
Ketones	-	76.26	148.58	127.63	149.612	190.8	523.12	375.31	170.57	1761.95
Acids	-	242.94	450.29	735.17	478.75	413.1	210.3	417.21	655.4	3603.16
Esters	20.64	555.53	178.68	381.4	1016.84	444.41	659.7	654.64	649.04	4560.88
Alcohols	25.88	1544.72	1699.12	2519.4	2142.66	1655.45	1038.01	1319.42	970.99	12,915.65
Hydrocarbons	-	696.99	659.16	1048.83	781.82	597.69	815.12	1012.76	611.6	6223.97
Others	23.25	1373.82	1778.92	2015.28	927.68	1504.21	1138.2	981.63	515.97	10,258.96
Total	171.13	10,714.79	12,042.53	17,355.65	10,984.48	9498.69	9362.33	11,862.63	5879.98	87,872.22

Note: - means the compound was not detected (the same below).

**Table 4 foods-13-02587-t004:** OAV values of volatile flavor compounds in different processing processes of Mianning ham.

Compounds	Threshold Value (μg/kg)	OAV Value (OVA ≥ 1)								
		HT1	HT2	HT3	HT4	HT5	HT6	HT7	HT8	HT9
Hexanal	7.5	4.48	409.86	459.84	671.5	345.3	181.33	115.49	337	111.62
Heptanal	10	0.69	34.21	32.95	47.18	22.47	18.8	24.78	33.1	9.27
Methional	0.04	-	-	-	-	-	-	525.33	807.55	334
Benzaldehyde	50	0.13	-	-	-	-	1.53	4.33	4.07	1.16
Octanal	0.1	148.04	5411.91	3805.37	6992.07	3188.86	3995.26	5535.58	5410.98	1268.06
Benzeneacetaldehyde	9	-	-	-	-	-	3.97	14.84	34.52	12.61
Nonanal	3.5	11.27	403.98	275.87	477.73	306.6	367.2	463.29	383.03	101.25
(E)-2-octenal	0.07	-	1577.11	3710.65	6882.78	2699.82	1707.69	1525.08	3295.86	780.82
Decanal	0.9	-	55.76	42.87	81.38	59.68	89.83	318.45	119.03	55.83
(E,E)-2,4-Nonadienal	0.06	-	-	1200.57	1830.18	663.47	389.68	212.68	852.52	137
Undecanal	14	-	-	-	-	-	0.82	1.82	1.3	0.29
(E,E)-2,4-Decadienal	0.03	-	-	1725.5	3698.97	1358.97	1208.43	1005.7	3887.7	379.63
Dodecanal	1.07	-	-	-	38.66	-	13.34	30.71	11.13	11.59
2-Heptanone	70	-	-	-	-	-	1	1.73	-	-
2-Nonanone	25	-	-	-	-	-	-	8.1	-	-
2-Undecanone	10	-	-	-	-	-	-	1.39	-	-
Hexanoic acid	200	-	0.21	1.48	2.63	1.34	0.37	0.05	0.24	0.11
1-Hexanol	200	-	0.24	0.16	0.32	0.13	1.06	0.7	0.34	0.37
1-Heptanol	200	-	0.57	0.54	0.95	0.42	0.61	0.62	0.66	-
1-Octen-3-ol	2	-	381.75	143.25	233.43	212.81	295.18	136.22	297.03	94.81
1-Octanol	54	-	3.52	-	-	-	4.24	-	-	-
Linalool	1.5	-	-	-	-	-	9.69	66.56	6.66	87.89
1-Nonanol	2	2.15	-	-	-	-	-	16.26	0	0
2-Pentyl-furan	4.8	-	17.74	16.51	25.73	19.21	17.03	42.63	36.78	21.26
2-Methoxy-phenol	0.17	-	-	-	-	-	-	-	-	124.76
Creosol	10	-	-	-	-	-	-	1.07	-	-
Estragole	7.5	-	7.99	6.39	-	-	-	18.67	3.54	12.32

## Data Availability

The original contributions presented in the study are included in the article/Appendix A, further inquiries can be directed to the corresponding author.

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
