# Peer review of "Study on the Changes and Correlation of Microorganisms and Flavor in Different Processing Stages of Mianning Ham"

_foods, 2024, doi:10.3390/foods13162587_

Round 1
Reviewer 1 Report
Comments and Suggestions for Authors
Methodology:
Were the same samples from the beginning through the final during 3 years? Do not explained during the text
It could not be concluded that the microorganisms at the beginning of maturation inhibit the growth of others during the following 3 years.
Results:
Table 1 and Table 2 are not necessary, only the Venn diagram
The rarefaction curve with the OTUS data, has to be explained
Appendix A Figure A1
Add as Supplementary
Author Response
Dear editor,
Thank you very much for taking the time to review this manuscript. I greatly appreciate your feedback and suggestions. We have carefully considered your comments and have made an effort to address each point. Below are our responses to your suggestions, and all modifications in the manuscript have been highlighted for your convenience.
Reviewer 1
Q1: Were the same samples from the beginning through the final during 3 years? Do not explained during the text
Response 1: We initially prepared a batch of ham raw materials and divided them into 9 groups according to the experimental design requirements. These groups were processed simultaneously in the factory. Subsequently, samples from each of the 9 batches were collected at various stages of processing according to the experimental design.
Q2: It could not be concluded that the microorganisms at the beginning of maturation inhibit the growth of others during the following 3 years.
Response 2: I fully accept your suggestions and have made the revisions in lines 239-242 of the manuscript.
Q3: Table 1 and Table 2 are not necessary, only the Venn diagram. The rarefaction curve with the OTUS data, has to be explained.
Response 3: I fully accept your suggestions and have removed Tables 1 and 2 from the manuscript. Additionally, I have included the supplementary explanation for the OTUs rarefaction curve in lines 169-173 of the text.
Kind regards,
Yue Huang
Reviewer 2 Report
Comments and Suggestions for Authors
Please consider: A total of 324 volatile compounds were identified, of which 27 were key contributors to the ham's flavor.
The text mentions that ….no such studies have been conducted on Mianning ham…. it would be beneficial to briefly mention why this gap in research is significant.
The purpose of the study is stated clearly, could it be more directly linked to practical outcomes?
Consider revising the sentence: The findings aim to provide a theoretical reference for improving the flavor quality of Mianning ham and enhancing its market competitiveness
to: This study investigates the dynamic changes in bacterial and fungal communities during the processing of Mianning ham and their impact on flavor compounds, aiming to provide a theoretical foundation for improving its flavor quality and market competitiveness.
This accumulation of hydrocarbons, exceeding that of ham from other regions in China, warrants further investigation.
The accumulation of hydrocarbons in Mianning ham, which surpasses that of hams from other regions in China, highlights a unique characteristic that warrants further investigation.
Comments on the Quality of English LanguagePlease consider: A total of 324 volatile compounds were identified, of which 27 were key contributors to the ham's flavor.
The text mentions that ….no such studies have been conducted on Mianning ham…. it would be beneficial to briefly mention why this gap in research is significant.
The purpose of the study is stated clearly, could it be more directly linked to practical outcomes?
Consider revising the sentence: The findings aim to provide a theoretical reference for improving the flavor quality of Mianning ham and enhancing its market competitiveness
to: This study investigates the dynamic changes in bacterial and fungal communities during the processing of Mianning ham and their impact on flavor compounds, aiming to provide a theoretical foundation for improving its flavor quality and market competitiveness.
This accumulation of hydrocarbons, exceeding that of ham from other regions in China, warrants further investigation.
The accumulation of hydrocarbons in Mianning ham, which surpasses that of hams from other regions in China, highlights a unique characteristic that warrants further investigation.
Author Response
Dear editor,
Thank you very much for taking the time to review this manuscript. I greatly appreciate your feedback and suggestions. We have carefully considered your comments and have made an effort to address each point. Below are our responses to your suggestions, and all modifications in the manuscript have been highlighted for your convenience.
Reviewer 2
Q1: Please consider: A total of 324 volatile compounds were identified, of which 27 were key contributors to the ham's flavor.
Response 1: I fully accept your suggestions and have made revisions in lines 20-21 of the manuscript.
Q2: The text mentions that ….no such studies have been conducted on Mianning ham…. it would be beneficial to briefly mention why this gap in research is significant. The purpose of the study is stated clearly, could it be more directly linked to practical outcomes?
Response 2: II fully accept your suggestions and have added a supplementary explanation on why similar theoretical research is necessary for Mianning ham in lines 67-71 of the manuscript, discussing its benefits.
Q3: Consider revising the sentence: The findings aim to provide a theoretical reference for improving the flavor quality of Mianning ham and enhancing its market competitiveness to: This study investigates the dynamic changes in bacterial and fungal communities during the processing of Mianning ham and their impact on flavor compounds, aiming to provide a theoretical foundation for improving its flavor quality and market competitiveness.
Response 3: Thank you for your suggestions. I have made the corresponding revisions in lines 75-78 of the manuscript.
Q4: This accumulation of hydrocarbons, exceeding that of ham from other regions in China, warrants further investigation.
The accumulation of hydrocarbons in Mianning ham, which surpasses that of hams from other regions in China, highlights a unique characteristic that warrants further investigation.
Response 4: Thank you for your suggestions. I have made partial revisions in lines 516-518 of the manuscript. Additionally, I have included the characteristic of Mianyang ham you mentioned as a new finding and a potential research target in lines 32-36 of the manuscript.
Kind regards,
Yue Huang